# Selecting Optimal Proton Pencil Beam Scanning Plan Parameters to Reduce Dose Discrepancy between Discrete Spot Plan and Continuous Scanning: A Proof-of-Concept Study

**DOI:** 10.3390/cancers15164084

**Published:** 2023-08-13

**Authors:** Xiaoying Liang, Chris J. Beltran, Chunbo Liu, Chunjoo Park, Bo Lu, Sridhar Yaddanapudi, Jun Tan, Keith M. Furutani

**Affiliations:** 1Department of Radiation Oncology, Mayo Clinic, Jacksonville, FL 32224, USA; 2Department of Radiation Oncology, The First Affiliated Hospital of Zhengzhou University, Zhengzhou 450052, China

**Keywords:** radiation oncology, particle therapy, pencil beam scanning, continuous scanning, plan parameters

## Abstract

**Simple Summary:**

Proton beam therapy delivered via pencil beam scanning is an advanced method for treating cancer. Two types of scanning modes are available for this method: discrete spot scanning (DSS) and continuous scanning. Continuous scanning has some advantages over DSS including improved proton beam delivery efficiency and reduced delivery time. However, continuous scanning delivers a dose between consecutive spots, and currently commercially available treatment planning systems do not account for this. Consequently, the continuous scanning delivered dose is inherently different from the planned one, which is discrete spot based. We conducted a proof-of-concept study of the ability of a prediction model to reduce the dose discrepancy between continuous scanning and discrete spot plan. Our findings suggest that selecting planning parameters with the prediction model reduces the dose discrepancy and therefore may permit relaxed dose delivery constraint, which further improves the benefits of continuous scanning.

**Abstract:**

Pencil beam scanning delivered with continuous scanning has several advantages over conventional discrete spot scanning. Such advantages include improved beam delivery efficiency and reduced beam delivery time. However, a move dose is delivered between consecutive spots with continuous scanning, and current treatment planning systems do not take this into account. Therefore, continuous scanning and discrete spot plans have an inherent dose discrepancy. Using the operating parameters of the state-of-the-art particle therapy system, we conducted a proof-of-concept study in which we systematically generated 28 plans for cubic targets with different combinations of plan parameters and simulated the dose discrepancies between continuous scanning and a planned one. A nomograph to guide the selection of plan parameters was developed to reduce the dose discrepancy. The effectiveness of the nomograph was evaluated with two clinical cases (one prostate and one liver). Plans with parameters guided by the nomograph decreased dose discrepancy than those used standard plan parameters. Specifically, the 2%/2 mm gamma passing rate increased from 96.3% to 100% for the prostate case and from 97.8% to 99.7% for the liver case. The CTV DVH root mean square error decreased from 2.2% to 0.2% for the prostate case and from 1.8% to 0.9% for the liver case. The decreased dose discrepancy may allow the relaxing of the delivery constraint for some cases, leading to greater benefits in continuous scanning. Further investigation is warranted.

## 1. Introduction

Cancer remains one of the most pressing challenges in modern medicine, affecting millions of lives worldwide [1]. Over the past decades, significant progress has been made in the development of various treatment modalities, and radiation therapy has proven to be a critical component in the management of cancer [2]. Particle therapy, a cutting-edge form of radiation therapy, has emerged as a promising avenue for the precise and effective treatment of cancer. Particle therapy employs charged particles, such as protons or heavy ions, to deliver radiation to tumor tissues. By taking advantage of the unique physical and biological properties of particles, this technique offers distinct advantages over traditional treatments, including reducing adverse events and allowing dose escalation to tumors [3,4,5,6,7,8,9].

Pencil beam scanning (PBS) is the primary modality in modern particle therapy facilities [10]. PBS provides better dose conformity to the target tissue than conventional passive scattering [11,12]. The theoretical basis for PBS techniques was largely formulated in the 1990s [13], in which both discrete spot scanning (DSS) and continuous scanning methods were proposed and investigated. The clinical use of PBS was initially implemented at the Paul Scherrer Institute [14] with DSS. DSS allows beam-monitoring systems to easily verify spot positions and the number of monitor units (MUs) delivered, which led to DSS becoming the primary scanning mode for proton PBS [15,16]. Continuous scanning was first implemented at the GSI [17] and is the main scanning mode in modern heavy ion therapy systems [18,19,20,21] and advanced proton therapy systems [22]. Continuous scanning has the potential to improve dose delivery efficiency and reduce beam delivery time (BDT) [23].

Recently, Hitachi made its continuous scanning mode, termed dose-driven continuous scanning (DDCS), also known as raster scanning, available for its proton PBS. To maintain consistency with Hitachi’s terminology, we will refer to it as DDCS throughout the remaining text. Our in silico studies have demonstrated that DDCS, compared to DSS, improves the irradiation duty factor from 14% to 41% [23] and reduces the average beam delivery time (BDT) by approximately 10% [24]. However, so far, no institutions with Hitachi proton therapy systems have implemented DDCS for clinical use. Our institution is preparing to install the newest Hitachi particle therapy system, which incorporates state-of-the-art advancements in technologies, including high scanning speed, high beam intensity, as well as proton PBS DDCS mode. However, before implementing this novel proton PBS scanning mode that utilizes the advanced machine operating parameters for clinical use, extensive studies are imperative.

A challenge of continuous scanning techniques, including DDCS, is that the move dose (the dose delivered during the transition from the previous spot position to the current spot position) is delivered between consecutive spots. In addition, current commercially available treatment planning systems (TPSs) perform only DSS-based optimization. Therefore, the move dose is not considered during treatment plan optimization, which results in an inherent dose discrepancy in continuous scanning. For particle therapy systems with continuous scanning, mitigation approaches that add constraints during delivery are used to achieve acceptable dose discrepancies [25,26]. One such approach involves defining an upper limit, as in reference [25], so that the beam is turned off when the distance between two consecutive spots exceeds the upper limit. For the Hitachi particle therapy system, a minimum stop ratio (SR) constraint of one is recommended by the vendor and is clinically applied at the Osaka Heavy Ion Therapy Center [26]. The minimum SR constraint guarantees a minimum ratio of the stop dose (the dose delivered at the spot position) to the move dose by limiting the beam current and designating some spots as break spots, which direct the beam to switch off and eliminate the delivery of the move dose between the previous spot and the break spot [24].

In our previous study [23], we simulated the DDCS dose discrepancy as a function of the minimum SR constraint. The results showed that an increase in the minimum SR constraint resulted in a decreased beam intensity and consequently led to longer BDT. While we found that the vendor-recommended minimum SR constraint of one yielded acceptable dose discrepancy for our studied cohort, we also observed that some plans had much smaller dose discrepancies than others. In some plans, acceptable dose discrepancies (<1%) were observed even with an extremely relaxed minimum SR constraint of zero. However, some plans showed 5% dose discrepancy [23]. This finding prompted us to investigate the factors that contribute to smaller dose discrepancies in some plans compared to others. Therefore, in the current investigation, we designed our study to answer the following questions: (1) Are there optimal plan parameters that lead to minimized DDCS dose discrepancy? (2) If so, can we develop a nomograph to guide the selection of these optimal planning parameters? The answers to these questions will help us further explore the practicality of DDCS.

## 2. Materials and Methods

### 2.1. Simulation of DDCS Dose Distribution and Evaluation of DDCS Dose Discrepancy

To simulate the dose distribution for DDCS, the DSS plan generated by the TPS needs to be converted for DDCS delivery. This conversion involves the utilization of two important concepts in DDCS simulation: the minimum SR constraint and break spots.

The SR is the ratio of the time spent at the spot position divided by the move time from the previous spot position to the current spot position. Therefore, the SR is equivalent to the number of MUs delivered at the stop position divided by the number of MUs delivered during the move time when a stable beam current is assumed.
(1)Stop ratioi,j=δtstopi,jδtmovei,j =MUspoti,jI−δtmovei,j δtmovei,j  
in which $\delta t_{move}^{i,j}\;$and $\delta t_{stop}^{i,j}$ denote the move time from the previous spot position to the current spot position and the time spent at the current spot position, respectively. $MU_{spot}^{i,j}\;$denotes the spot MU from discrete spot plan. The index of the energy layer is denoted by ***i***, and ***j*** is the index of the spot in each layer. ***I*** denotes the beam current expressed as the number of MUs per second.

The minimum SR constraint is a technique used in the Hitachi particle therapy system to control the move dose contribution and, consequently, to control the DDCS dose discrepancy from the planned discrete spot dosimetry. Equation (1) shows that lowering the beam current is needed to meet the minimum SR constraint; thus, a larger minimum SR constraint indicates that a lower beam current is required. In Hitachi systems, only 1 beam current is applied to a given layer. Therefore, when the number of MUs for a given spot is small, the beam current must be reduced to a low value to meet the minimum SR constraint for that spot, and this low beam current must be applied to all spots in the layer. This, in turn, leads to a prolonged BDT [24]. To address this issue, break spots which are also referred to as break points (BP) are introduced. Because the beam is switched off when moving to a BP position, no move dose is delivered for BPs, and their SR is infinitely large. By designating spots with a small number of MUs to BPs, a reasonable beam current may be used while still meeting the minimum SR constraint.

For DDCS delivery, the optimized beam current for each layer depends on both the minimum SR constraint and the determination of break spots. We used a beam current threshold of 10 MU/s for the determination of break spots according to our previous study [24]. Subsequently, the beam current for each energy layer (ranging from 10 to 30 MU/s) was optimized to achieve both the fastest BDT and ensure that each non-break spot meets the minimum SR constraint. The dose discrepancy of delivering a discrete spot plan with DDCS using minimum SR constraints of 1 and 0 was simulated. For simulations with a minimum SR constraint of 0, only spots whose move MU exceeded the spot MU were designated as BPs.

The DDCS simulations utilized the machine operating parameters of our institution’s upcoming particle therapy system, including a maximum beam current of 30 MU/s and scanning speeds of 20 m/s and 8 m/s in the y- and x-directions, respectively. With a typical spot spacing of 5 mm for proton plans, the scan time between consecutive spots ranged in the order of a few hundred microseconds. In our simulation, we made the assumption that the action of the scanning magnet was linear.

Detailed information on the DDCS dose simulation has been previously described [23]. Briefly, a DDCS spots map file was generated by using an in-house program. In the DDCS spots map, 10 auxiliary spots were inserted between consecutive spots to represent the move dose. The DDCS spots map was then imported back to the Eclipse TPS for DDCS dose calculation. The dose discrepancy was evaluated by using gamma analysis and the CTV dose-volume histogram (DVH) percentage root mean square error (RMSE). Because the DDCS dose was simulated with the same computed tomography image sets used for planning, no setup error occurred. Therefore, we performed gamma analysis with a more stringent criteria of 2%/2 mm. The gamma analysis was conducted using 10% of the maximum as the cutoff threshold. The CTV DVH RMSE was calculated as:(2)$$RMSE=\sqrt{\frac{\sum_{i=1}^{n}{\left[\left(D1_{i}-D2_{i}\right)/D1_{i}\right]}^{2}}{n}\;\;\;\;\;\;\;\;\;}\;$$

***D*1** and ***D*2** represent the dose values from the DSS plan and the DDCS simulation, respectively. The CTV volume percentile is denoted by ***i***, which ranged from 1 to 100, with a step size of 1. ***D_i_*** represents the dose that was received by at least ***i***% of the volume.

### 2.2. DSS Plans with a Cubic Target

Plans were created using the Eclipse TPS, v15.6 (Varian Medical Systems) employing an anterior-posterior beam for a cubic target with a volume of 3 × 3 × 3 cm^3^ in a water phantom, with the center of the target located at a depth of 15 cm. Radiation dose levels of 0.5, 2.0, 5.0, and 10.0 GyRBE (RBE = 1.1) were chosen to include the range of prescriptions used in clinical practice, with consideration that one to four beams are typically used in a plan. For each dose level, multiple plans were generated with different combinations of minimum MU/spot and spot spacing. We used three distinct minimum MU/spot (0.1, 3.0, and 6.0 milli MU) and three spot spacings (0.5, 1.0, and 1.5 sigma). In our clinical experience, a minimum MU/spot of 3 milli MU and a spot spacing of 1.0 sigma are typically used for planning. In our system, 1 MU corresponds to approximately 0.9 billion protons at 230 MeV and approximately 0.4 billion protons at 70 MeV [27]. 

### 2.3. Prediction Model (Nomograph) Generation

We hypothesized that an optimal combination of minimum MU/spot, spot spacing, and field dose can yield a desired spot pattern and thereby reduce the DDCS dose discrepancy. Therefore, a multivariable regression was performed with the simulated results of these 28 plans by using minimum MU/spot ($x_{1},\;mill\;MU)$, spot spacing ($x_{2},\;sigma)$, and field dose ($x_{3},\;GyRBE)\;$as predictors and CTV DVH RMSE (y) as the response. The regression analysis was performed with MATLAB software (version R2019b MathWorks) [28] using quadratic fitting. The decision of using quadratic fitting was derived from intuition about the interrelation between the Minimum MU/spot, spot spacing, and field dose. The response function y is described as the trace of scaler product between fitted coefficients matrix A and correlation matrix of predictors as follows:(3)$$y=Tr\left(A\cdot \hat{X}\right)=Tr\left(A\cdot \left(X^{T}X\right)\right)$$
$$\mathrm{which}\;\;\;\;\;\;\;\;A=\left[\begin{array}{cc} \begin{array}{cc} a_{0} & a_{1}\\ 0 & a_{4} \end{array} & \begin{array}{cc} a_{2} & a_{3}\\ a_{5} & a_{6} \end{array}\\ \begin{array}{cc} 0 & 0\\ 0 & 0 \end{array} & \begin{array}{cc} a_{7} & a_{8}\\ 0 & a_{9} \end{array} \end{array}\right]\;$$
and $X=\left[1,\;x_{1},\;x_{2},x_{3}\right]$ is the vector of predictors. $\;x_{1},\;x_{2},x_{3}$ represent the Minimum MU/spot in mill MU, spot spacing in sigma, and field dose in GyRBE.

The purpose of the prediction model was not for precise prediction of the dose discrepancy but rather to provide a nomograph-type guide to select planning parameters of spot spacing and minimum MU/spot according to prescription (field dose). Therefore, we categorized the predicted risk of dose discrepancy as specific levels for predicted CTV DVH RMSE values less than 0.5% (level-1), from 0.5% to 1.0% (level-2), from 1.0% to 1.5% (level-3), and from 1.5% to 2.0% (level-4).

### 2.4. Model Testing with Clinical Cases

Two clinical cases, one prostate (CTV volume, 55 cm^3^; prescription, 2.5 GyRBE/fraction) and one liver (CTV volume, 100 cm^3^; prescription, 1.8 GyRBE/fraction), were used to test the model. The clinical plan with our commonly used plan parameters (minimum MU/spot of 3.0 milli MU and 1 sigma spot spacing) was used as the baseline. For each case, the predicted DDCS dose discrepancy level was calculated first by using the clinical plan parameters. If the predicted dose discrepancy level for the clinical plan exceeded 1, we made adjustments to the spot spacing and/or the minimum MU/spot. In the Eclipse TPS, the spot spacing is defined at the plan level, while the minimum MU/spot is defined at the machine level. Changing the spot spacing is a relatively straightforward process, involving a modification to a single parameter within the plan. On the other hand, altering the minimum MU/spot requires a new machine model. Therefore, when the predicted dose discrepancy level for delivering the clinical plan with DDCS exceeded 1, our initial approach was to modify the spot spacing to either 0.5 sigma or 1.50 sigma while keeping the minimum MU/spot at 3.0 milli MU. If the predicted dose discrepancy level remained higher than 1, we then adjusted the minimum MU/spot until the predicted dose discrepancy level was reduced to 1. After the combination of minimum MU/spot and spot spacing to achieve level 1 dose discrepancy prediction was identified, a test plan with these parameters was generated. The clinical and test plans used the same beam angles and cost function. DDCS with a minimum SR constraint of 0 was then simulated for both the clinical plan and the test plan. The simulated DDCS dose discrepancy was then compared between the clinical plan and the test plan to evaluate the effectiveness of the nomograph model.

## 3. Results

### 3.1. DDCS Simulation of Treatment Plans for Cubic Targets

Figure 1 displays the axial view of the dose distributions for the nine plans utilizing different combinations of minimum MU/spot and spot spacing, all with a dose of 2 GyRBE. Comparable dose distribution was achieved.

When a minimum MU/spot of 6.0 milli MU was used to generate plans with a 0.5 GyRBE dose, only the plan with a spot spacing of 1 sigma successfully achieved 95% coverage to the CTV. Therefore, we conclude that the spot spacing of 0.5 sigma and 1.5 sigma, together with the minimum MU/spot of 6.0 mill MU, are not appropriate planning parameters for generating 0.5 GyRBE plans. Consequently, a total of seven plans were generated with a prescription of 0.5 GyRBE. 

Based on the simulation results of the 0.5 GyRBE plans and the 2 GyRBE plans (Appendix A), we observed that plans with a minimum MU/spot of 0.1 milli MU had a higher fraction of BPs compared to plans with minimum MU/spot of 3 milli MU or 6 milli MU. Additionally, we noted a smaller fraction of BPs when employing a minimum SR constraint of 0 for DDCS simulation, compared to using a minimum SR constraint of 1. A high fraction of BPs defeats the purpose of DDCS. Therefore, for plans with doses of 5.0 GyRBE and 10.0 GyRBE, minimum MU/spot values of 3.0 milli MU or 6.0 milli MU were utilized, and DDCS was simulated exclusively with a minimum SR constraint of 0. 

The DDCS dose discrepancies (CTV DVH RMSE and 2%/2 mm gamma passing rate) for all simulations are summarized in Appendix A. The fraction of BPs and 1%/1 mm gamma passing rate are also shown. 

When utilizing a minimum SR constraint of 0 for DDCS simulation, the 0.5 GyRBE plans with a minimum MU/spot of 6 mMU had a CTV DVH RMSE of 4.5% and a 2%/2 mm gamma passing rate of 81.4%. Among the remaining six plans with 0.5 GyRBE dose, four plans had a CTV DVH RMSE of 1.0% or lower and a 2%/2 mm gamma passing rate of 98.0% or higher. For the nine plans with 2.0 GyRBE dose, seven plans achieved a CTV DVH RMSE of less than 1.0% and a 2%/2 mm gamma passing rate greater than 99%. For all plans with 5.0 GyRBE and 10.0 GyRBE doses, the CTV DVH RMSE was less than 1.0%. The 2%/2 mm gamma passing rates were 100.0% for all these plans, except for the 5.0 GyRBE plan that used a minimum MU/spot of 6 milli MU and a spot spacing of 0.5 sigma, which had a 2%/2 mm gamma passing rate of 97.9%.

When a minimum SR 0 was used for DDCS simulation, for the 0.5 GyRBE plans with minimum MU/spot of 6 mMU, the CTV DVH RMSE was 4.5% and the 2%/2 mm gamma passing rates was 81.4%. For the other six plans with 0.5 GyRBE dose, four of the six plans had 1.0% or lower CTV DVH RMSE and 98.0% or higher 2%/2 mm gamma passing rate. For the nine treatment plans with a 2.0 GyRBE dose, seven of the nine treatment plans had less than 1.0% CTV DVH RMSE and greater than 99% 2%/2 mm gamma passing rate. All plans with 5.0- and 10.0-GyRBE doses had less than 1.0% CTV DVH RMSE. The 2%/2 mm gamma passing rates were 100.0% for all of these plans, except for the 5.0-GyRBE plan that used a minimum MU/spot of 6 milli MU and spot spacing of 0.5 sigma, which had a 2%/2 mm gamma passing rate of 97.9%.

To understand the underlying reason for varied DDCS dose discrepancies among plans with the same dose and comparable dose distribution, we plotted spot patterns at the center of the spread-out Bragg peak for representative treatment plans with a 2.0 GyRBE (Figure 2). Although all representative treatment plans had comparable dose distributions, the spot patterns varied widely.

### 3.2. Nomograph Prediction Model

The multivariate regression fitting yielded a coefficient of determination (R^2^) of 0.8. The fitting coefficients are listed in Table 1. The visualization of the fitting is displayed in Figure 3, utilizing the partial regression plot [29] generated by MATLAB. This plot employs the Frisch–Waugh–Lovell theorem [30,31] to partial out the variables with the dependent variable also partialled out in the same way.

### 3.3. Model Testing with Clinical Cases

Figure 4 shows the model-predicted DDCS dose discrepancy levels for the prostate case and the liver case. The figure displays the predicted dose discrepancy levels for the clinical planning parameters (minimum MU/spot of 3 milli MU and spot spacing of 1 sigma), as well as various other combinations of the planning parameters. As can be seen in Figure 3, a level-3 dose discrepancy for the prostate case using clinical plan parameters was predicted. When we adjusted the spot spacings to 0.5 sigma and 1.5 sigma, the model predicted level-3 and level-1 dose discrepancies, respectively. Consequently, we developed a test plan with a minimum MU/spot of 3 milli MU and 1.5 sigma spot spacing. For the liver case, the model also predicted a level-3 dose discrepancy with clinical plan parameters. After we adjusted the spot spacing to 0.5 sigma and 1.5 sigma, the model predicted level-4 and level-2 dose discrepancies, respectively. We then changed the minimum MU/spot to 1 milli MU, which resulted in level-2, level-2, and level-1 dose discrepancies for spot spacings of 0.5 sigma, 1.0 sigma, and 1.5 sigma, respectively. We therefore created a test plan with a minimum MU/spot of 1.0 milli MU and 1.5 sigma spot spacing for the liver case.

DDCS simulations using a minimum SR of 0 were subsequently performed for both the clinical plan and the test plan. The results are summarized in Table 2. The test plans that were developed with the model-guided planning parameters resulted in lower dose discrepancies than did the clinical plans for both cases. An earlier study showed the benefit of BDT for DDCS [24]. To ensure that our approach dose did not increase the BDT, we conducted BDT simulations on both the clinical plans and the test plans using our in-house program [24]. The BDT time (Table 2) was found to be comparable between the clinical plan and the test plan for both cases.

## 4. Discussion

DDCS is an irradiation technique of proton therapy that is still little studied and has much potential for development. In comparison with DSS, DDCS offers several benefits in PBS beam delivery, which includes improved irradiation duty factor and reduced beam delivery time [23,24]. However, DDCS delivery with DSS-optimized plans may result in dose discrepancies. In this study, we conducted an in silico investigation using the beam delivery parameters of our upcoming Hitachi particle therapy system. Our primary focus was to explore a potential method for reducing the DDCS dose discrepancy from DSS-optimized plans. It is important to highlight that this dose discrepancy is not specific to Hitachi systems; rather, it is a common challenge encountered with any particle therapy system that delivers continuous beams, due to the inherent difference from the discrete spot nature of the TPS.

In this study, using systematically planned cubical plans, we demonstrated that there is an optimal combination of minimum MU/spot, spot spacing, and field dose which can achieve a desired spot pattern that leads to minimized DDCS dose discrepancy. We developed a prediction model that may serve as a nomograph to guide the selection of the plan parameters (minimum MU/spot and spot spacing) according to the field dose. We then tested the effectiveness of the model with two clinical cases. By choosing the plan parameters guided by nomograph, we showed that the test plans yield lower DDCS dose discrepancy than the plans using typical clinical practice parameters. At the same time, we would also like to point out that the dose discrepancy level predicted by the nomograph model may exhibit slight variations from the actual dose discrepancy level. For instance, while the nomograph predicted level-3 for the prostate clinical plan, the actual simulation yielded level-4. Nevertheless, the nomograph demonstrated its capability in assisting with the selection of plan parameters, resulting in reduced dose discrepancies.

The reduction in BDT is a promising advantage for DDCS. While the plan parameters guided by the nomograph resulted in lower dose discrepancies, it is important to ensure that this approach does not lead to an increase in BDT. Our BDT time simulations demonstrated comparable results between the clinical plan and the test plan in both cases. In each case, the spill change time and energy layer switch time are essentially the same for the clinical plan and the test plan, and they contribute significantly to the total BDT. Specifically, the spill change time and the energy layer switch time occupied over 70% and 80% of the total BDT for the prostate case and the liver case, respectively. Consequently, the variations in the spot pattern between the clinical plan and the test plan only had a minor impact on the total BDT. 

Because this investigation was a proof-of-concept study, the nomograph model we generated was based on plans for cubic targets in water phantoms, with a uniform dose distribution. Therefore, we selected clinical cases that used single-field optimization as testing cases. In real clinical scenarios, many treatment sites, such as the head and neck, require multifield optimization. Therefore, to fully gauge the clinical benefit of our concept, the planning of technique-specific or even site-specific models may be required, and additional more extensive studies are necessary. In addition, the dose distribution of DDCS is influenced by beam delivery parameters such as scanning magnet scan speed and beam intensity. Therefore, a different model may be required for delivery systems with different beam delivery parameters. 

We would like to acknowledge that this investigation is an in silico simulation study. Once our particle therapy system is operational, it is essential to conduct measurements to validate our simulation results. Another limitation of this study is the manual process involved in selecting the ‘trial’ parameters used to feed into the nomograph. Future works on the development and utilization of proper searching algorithms could enable a more extensive exploration of the potential of our proposed concept.

The main reason for the issue of DDCS dose discrepancy is that the current TPSs can only optimize DSS plans. The mitigation method proposed in this study has potential for reducing the dose discrepancy. However, a fundamental solution would be the direct optimization of a continuous beam within the TPS. Pioneering research by Trofimov and Bortfeld [32] found that continuous delivery dose discrepancies can be reduced by iteratively optimizing the simulated continuous beam fluence profiles during treatment planning. In general, direct optimization for a continuous beam would require a significantly larger solution space compared to the current TPS optimization approach. One application of our method is to use the results of the nomograph as the initial variable setting for a DDCS-based TPS planning optimization scheme. This reduces the solution space and potentially leads to improved convergence rates compared to random parameter selection. Starting from a nearly optimal point increases optimization efficiency and decreases the likelihood of getting trapped in a local optimum [33].

## 5. Conclusions

An optimal combination of minimum MU/spot and spot spacing according to field dose may lead to a desired spot pattern and yield low DDCS dose discrepancy. We developed a prediction model to serve as a nomograph for selecting plan parameters. The effectiveness of this approach was confirmed by testing with prostate and liver clinical cases. Our findings have the potential to reduce DDCS dose discrepancy. The reduced DDCS dose discrepancy may allow for a relaxed delivery constraint for some cases, which will thereby lead to greater benefits of DDCS. Further investigation is necessary to fully evaluate the benefits of this approach.

## Figures and Tables

**Figure 1 cancers-15-04084-f001:**
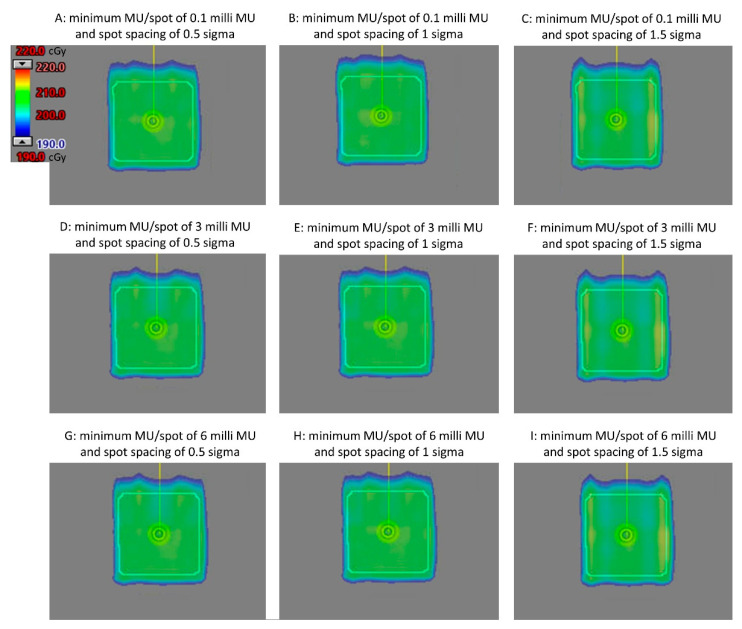
Axial view of the dose distribution for plans using various planning parameters with a 2.0 GyRBE dose prescription, using an anterior-posterior beam.

**Figure 2 cancers-15-04084-f002:**
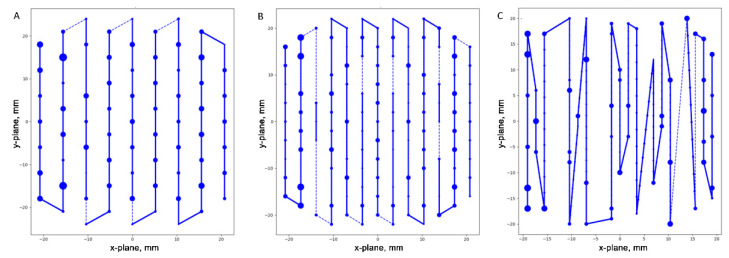
Spot pattern at center of spread-out Bragg peak for representative treatment plans with 2.0 GyRBE. (**A**) Plan with minimum MU/spot of 0.1 milli MU and spot spacing of 1.5 sigma. (**B**) Plan with minimum MU/spot of 3 milli MU and 1.0 sigma spot spacing. (**C**) Plan with minimum MU/spot of 6 milli MU and 0.5 sigma spot spacing. Size of the dots is proportional to the MU. Lines depict the scanning paths: solid lines indicate continuous scanning and dashed lines indicate break spots.

**Figure 3 cancers-15-04084-f003:**
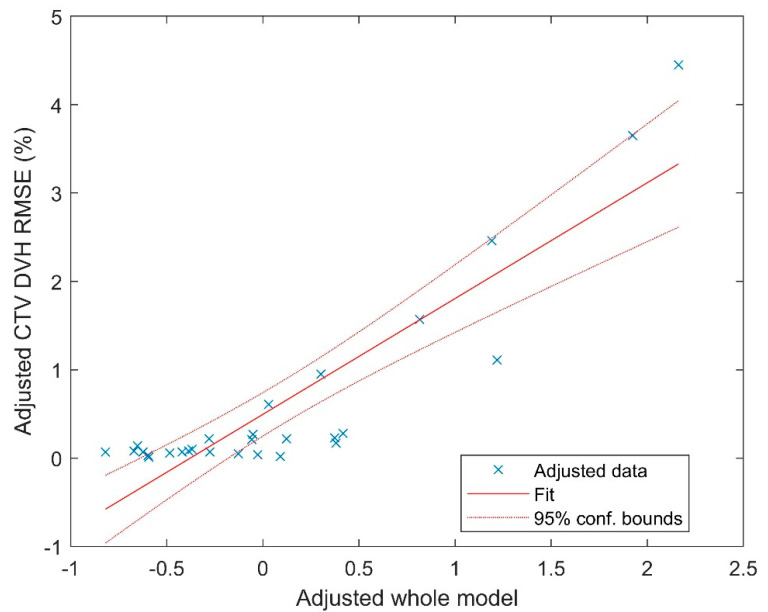
Partial regression plot to visualize the multivariate regression fitting.

**Figure 4 cancers-15-04084-f004:**
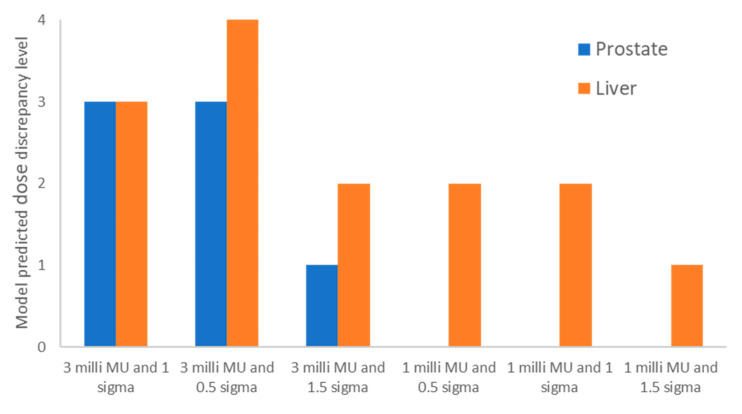
Model-predicted DDCS dose discrepancy levels for the prostate and liver cases using different planning parameters, including the clinical plan parameters (minimum MU/spot of 3 milli MU and spot spacing of 1 sigma) and other combinations. In the x-axis label, the values indicate the minimum MU/spot and spot spacing.

**Table 1 cancers-15-04084-t001:** Regression coefficients.

Coefficients	$a_{0}$	$a_{1}$	$a_{2}$	$a_{3}$	$a_{4}$	$a_{5}$	$a_{6}$	$a_{7}$	$a_{8}$	$a_{9}$
Values	0.496	0.659	0.634	−0.737	0.040	−0.390	−0.073	−0.365	0.202	0.059

**Table 2 cancers-15-04084-t002:** Prediction model testing results with clinical cases.

Clinical Case	Dose/Fraction, GyRBE	Plan Type	Minimum MU/Spot, milli MU	Spot Spacing, sigma	Field Dose, GyRBE	Predicted Dose Discrepancy Level	DDCS CTV DVH RMSE, %	DDCS 2%/2 mm gamma Passing Rate, %	Beam Delivery Time, s
Prostate	2.5	Clinical	3.0	1.0	1.25	3	2.2	96.3	34
		Test	3.0	1.5	1.25	1	0.2	100.0	32
Liver	1.8	Clinical	3.0	1.0	0.6	3	1.8	97.8	50
		Test	1.0	1.5	0.6	1	0.9	99.7	51

Abbreviations: CTV, clinical target volume; DDCS, dose-driven continuous scanning; DVH, dose-volume histogram; GyRBE, relative biological effectiveness of physical dose in Gray; MU, monitor unit; RMSE, root mean square error.

## Data Availability

The data that support the findings of this study are available from the corresponding author upon reasonable request.

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
