# Peer review of "Selecting Optimal Proton Pencil Beam Scanning Plan Parameters to Reduce Dose Discrepancy between Discrete Spot Plan and Continuous Scanning: A Proof-of-Concept Study"

_cancers, 2023, doi:10.3390/cancers15164084_

Round 1

Reviewer 1 Report

Summary

This is the original version of a manuscript submitted to and reviewed for Cancers. The current manuscript describes a proof of concept for reducing dose discrepancy between the DSS plan and the DDCS plan. The nomograph generated from the phantom study was created to select planning parameters. The authors found a reduced discrepancy in the clinical cases with the selected planning parameters from the monograph.

General Comments

As noted, this is the original version of a manuscript submitted to and reviewed for Cancers. DDCS is attracted in terms of the advantage of dose delivery and accelerator construction. However, current commercially available TPSs perform only DSS-based optimization. Therefore, move dose is not considered during treatment plan optimization. There are many combinations of the adjustable planning parameters such as minimum MU/post, spot spacing, and dose per beam. The authors suggested using a nomograph for guidance to select planning parameters to reduce the discrepancy between the DSS plan and the DDCS plan. The concept is useful for our community. The reviewer found the following questions to be further considered in this journal.

Note that the number of pages and lines in this review refers to the number written in the clean copy version of the manuscript.

1) The authors should show the impact of the nomogram on the repainting plan.

The repainting technique is used to mitigate motion-related dosimetric error for moving targets in particle therapy. The minimum MU/spot is strongly related to the repainting number according to the prescription dose in the repainting technique. The spot spacing has also an impact on the flatness of the dose distribution. At least one of the clinical cases, i.e. liver case should show the availability of the concept for the repainting plan.

Specific Comments

Abstract Some concrete values (e.g. shown in the results) should be included in the abstract.

P2 line48 “This technique offers distinct advantages over traditional treatments.” Please write a few examples of the distinct advantages.

P4 line167 The treatment plan should be called as DSS plan through the text.

P4 line181 How did the authors determine the number of the coefficient (here a0 to a9) to fit? Is there any optimization process to increase the number of coefficients? You have 4 predictors, and the number of coefficients is supposed to be 4.

P5 Figure1 Values of minimum MU/post, spot spacing, and dose per beam written in the caption should be included in the figure as well. It would make the figure clearer.

P5 Figure1 It should indicate which plane is shown in the figure. Is it a coronal plane?

P8 Figure2 Please explain the dot size difference in the figure. Is it proportional to the dose of the point?

P8 Figure3 What does the “adjusted” mean? It seems to come from MATLAB function (plotAdded?). If so, the function you used to fit should be written at least for reference.

P8 Figure3 Please explain the meaning of the horizontal axis.

P8-9 278-283 It would be valuable to plot the points of the clinical cases that the authors predicted with the nomograph.

Reviewer 2 Report

In this manuscript, the authors report on a study to optimize irradiation parameters for continuous beam scanning in proton therapy. In contrast to discrete spot scanning, continuous scanning can be more efficient and reduce irradiation times. Since therapy planning systems currently only handle discrete scanning, there are dose differences if continuous scanning is applied directly. In their study, the authors developed a model to estimate the expected dose discrepancy for a selected set of irradiation parameters. The model was developed on a rather unrealistic case of a cubic dose distribution, but at least tested on two real cases.

Continuous scanning is an irradiation technique of proton therapy that is still little studied and has much potential for development. Moreover, little literature is available on it so far. Therefore, contributions in this field are welcome. The presented method shows the correlation of the different irradiation parameters and can provide further input for the readership. However, the developed model can only provide an indication of the expected dose difference. Moreover, the model is strongly dependent on the other accelerator parameters and cannot be directly transferred to other facilities. Furthermore, as the authors themselves mention, the model is likely to vary for different tumour types or sizes. Therefore, the direct benefit for the reader remains limitted.

General comments [G]:

G1: It is not entirely clear which irradiation parameters were used for the clinical case. And also DSS or DDCS? In the introduction, the authors write that none of the  Hitachi facility uses DDCS clinically, but in Table 3, the clinic plan is shown as DDCS.

G2: In order to understand better the differences between DCS and DDCS, it would be helpful if the authors could indicate typical times of the scanning process, e.g. the time for moving from one spot to the next or the minimum spot duration in ms and not only in mMU

G3: The authors call their scanning methods DDCS.  However, when this technique was introduced at GSI it was called ‘raster scanning ‘and this has remained an established term to this day. Is there any particular justification for this? I am not suggesting a change of name, but a reference to this term should be given.

G4: What is commonly called ‘field’, the authors refer to as ‘beam’ (e.g. line 104 or line 186 dose/beam -> field dose). Is there a specific reason for this?

G4: I would propose to use ‘ms’ instead of ‘msec’

Specific comments [S]:

S1, line 60: ‘Continuous scanning puts less wear on accelerator magnets’. For the magnet itself, it does not matter if the field changes with a high frequency. However, for the power supplies of the involved magnets, often and fast switching is an additional burden.
On the other hand, continuous scanning also places stricter demands on the magnets of the scanner.

S2, l87: ‘slower’ -> ‘lower’ or probably ‘higher’?

S3, l89: Would ‘SR constraint >=1’ not be more helpful for the understanding?

S4, l106: mMU is used, and not all readers may be familiar with the abbreviation milli MU. Listing it at the end of the text as an abbreviation may not be sufficient, especially as it appears together with minimum MU at this point.

S5, l109: The reviewer appreciate that 1 MU is defined. This makes the study more transferable to other institutions.

S6, l118: ‘filtered’: how and what was the criteria for filtering the plans?

S7, l148: ‘threshold of10 mMU/ms’: Not fully clear, does this mean that the beam current was change between 10 and 30 mMU/ms?

S8, l159: ‘auxiliary spots’: How many additional spots were introduced between two consecutive spots. On a fixed spacing or a fixed number?
Finally, also the action of the scanner magnet is not linear due to the characteristics of the power supply. Was this fact considered in the simulation?

Figure 1: Legend doesn’t have units. The grey background is not needed and should be removed. Instead of the extensive figure caption, each plots could be labelled with ‘0.1mMU, 1s’, etc.

Table 1: I somewhat question the usefulness of this table and wonder if the information could be summarized in a different way.

Figure 3: Difficult to extract any information from this graph. Label and title: What is ‘adjusted’ CTV, ‘adjusted whole model’, ‘added variable plot’? Correspond the data points below 0.5% to level 1 dose discrepancy? Caption also needs revision.

S9, l179: ‘level-4’ not defined.

S10, l187: ‘does’ -> does and ‘in-crease’ -> increase

S11, l341: ‘this study was a proof-of-concept exercise’ -> this investigation was a proof-of-concept study’

S12, l380: Add also BDT to abbreviations

S13, l381: Replace ‘:’ by ‘,’

For this reviewer, as a non-native speaker, the quality of the language is good. A few minor errors require a careful read-through of the entire manuscript.

Reviewer 3 Report

The manuscript describes a prediction model for proton pencil beam scanning (PBS) to minimize the dose discrepancy between the treatment planning system (TPS), which models discrete spot patterns, and the actual delivered dose in continuous scanning mode, which delivers dose when moving from one spot to the next. The presentation of the work is generallyclear; however, there are major concerns that need to be addressed.

Context of the work. This intrinsic discrepancy between the actual continuous scanning delivery and the discrete spot modeling in the TPS is not unique to the Hitachi system. For instance, the Varian ProBeam system also has a similar feature. While I agree with the authors that this discrepancy is intrinsic and interesting to investigate, it is not clearly laid out in the manuscript whether the dosimetric consequences due to this discrepancy are clinically significant, by itself without any correction, or after using currently available solutions. What levels of discrepancy might one expect (e.g., 1%? 5%?)? And in what clinical scenarios? It is very important to put this work in such a context to highlight the value and contribution of this work and for the readers to understand and judge the significance of such work. Without such a concrete context, the value of such an investigation would remain mostly in a “theoretical” sense.

Novelty of the work. While it is totally acceptable to investigate this intrinsic discrepancy from a more theoretical/methodological sense, the novelty of this proposed prediction model itself needs further justification/development. In this work, a rather simple regression model with three input parameters (minimum MU/spot, spot spacing, and dose per beam) was built. As the core part of the work, it is rather weak. Not that there is anything intrinsically wrong with simple models, it is more that there is no discussion as to why this particular model was chosen (e.g., were other models tried but not as good?)? What were the other options? While the authors did point out that the purpose of this model was not for precise prediction of the dose discrepancy but for establishing a monograph, it was not clear why this decision was made. That is, why not build a more precise model that could predict the dose discrepancy more accurately?

Impact and clinical usage of the work. While I do understand that the authors argue this work is intended to be a “proof-of-concept” study, I still think it is important to demonstrate, even as a proof-of-concept, to a sufficient extent such that it is potentially clinically useful. With the only
two clinical cases that were used for the testing in this work, one would hardly be convinced. In addition, the test was only restricted to SFO, while clinical MFO cases are not tested. MFO techniques are widely used for many disease sites as the authors pointed out, and arguably one of the most important reasons that makes PBS appealing. Given that, it was not clear how this study can translate to a MFO context. If the authors think it can only be restricted to SFO, then more SFO clinical cases are needed to really show that it’s robust in various real clinical settings. If the authors think it can be generalized to MFO, then additional evidence should be required. Without either of them, the value of such a “proof-of-concept” would be quite questionable.

Reviewer 4 Report

This is an interesting proof of concept study to reduce the discrepancy of continuous scanning delivered dose and planned dose which is discrete spot based. Studies to operate commercially available systems, such as treatment planning system and beam delivery system, for clinical use are important, and this manuscript will help users. But the contents of sections are a bit mixed up and it might be difficult to understand this manuscript on first reading readers. Also, the process of how parameters were guided by the nomograph is unclear. There is no obvious nomograph which derive the optimal combination parameters, but tables with a list of numerical values. Modifications are required to make the story strait and clear.

General comments:

P2, L65  The word of “irradiation duty factor” suddenly appeared. You should add short explanation of how duty factor was improved to help the reader. Or put memo like “see section XX”, etc.

P2, L82 The words of “stop dose” and “move dose” should be defined when they are first use in this manuscript.

P3, L113 “When a minimum MU/spot of 6.0 mMU was used to generate treatment plans with a 0.5 GyRBE dose, only the plan with 1 sigma spot spacing had reasonable coverage”. This should be described in section of Results? Consider chapter setting and that contents. Or, explain the criteria of acceptable plan of your prescription for clinical use, here.

P3, L118 “We then filtered a useful subset of planning parameters according to the DDCS simulation results with the 0.5 and 2.0 GyRBE doses. With this filtered subset of planning parameters…” How did you filter planning parameters? Process should be described, in here.

P4, L172

Please unify the terms used. dose per beam -> dose/beam?

P5, L188 level 1, level 2 and level 3 -> level-1, level-2 and level-3

P5, L195 “If the predicted dose discrepancy level for the clinical plan exceeded 1, we adjusted the spot spacing to either 0.5 sigma or 1.0 sigma while maintaining the minimum MU/spot at 3.0 mMU.”

You conducted a proof-of-concept study of the ability of a prediction model. So, can you show us the process, which clinical plan exceeded 1? 

P5, L200 “We adjusted the minimum MU/spot last because the Eclipse TPS required a new machine model to change it, whereas the spot spacing could be changed easily at the plan level.” 

Eclipse used can understand, however, non-user may not be able to understand it. Short explanation should be added.

P5, L191-205  2.5 Model Testing With clinical Cases.

Two clinical cases were generated with SR constraint of 1, as vender recommendation? Please briefly describe the reason why test plan was generated with SR constraint of 0, not 1. Or, can you show the results of test plan with SR constraint of 1?

P7, L235 “reasonable plan quality with 95% CTV covered by the prescription dose”. What reasonable plan exactly means? Gamma passing rate with 81.4% is still reasonable for clinical plan? How unreasonable the plans with another spot spacing 0.5 and 1.5 were? Could you briefly explain the reason why spot spacing of 0.5 and 1.5 sigma are not effective to generate clinical plan?

P7, L247 “Therefore, minimum MU/spot of 3.0 mMU or 6.0 mMU were used for treatment plans with 5.0- and 10.0-GyRBE doses...”

According to the story of this manuscript, you observed a smaller fraction of break spots with a SR constraint of 0, and the treatment plans with minimum MU/spot of 0.1mMU had a higher fraction of BPs than plans with 3mMU or 6mMU minimum MU/spot. But, minimum MU/spot of 0.1mMU were not used in the first place. Consider to make the story straight from 2. Material and Method to 3. Results.

P8, 3.2. Nomograph Prediction Model. Short explanation may be needed about what you got as a result of the multivariate regression analysis of the plans. What is the definition of “Adjusted data” and “Adjusted whole model” in figure 3? What is Horizontal axis? Whole explanatory variable? 

P9, L281 “level-2” is repeated.

P8, L273 – P9, L283 “When we adjusted the spot spacings to 0.5 sigma ...... therefore created a test plan with a minimum MU/spot of 1.0 mMU and 1.5 sigma spot spacing for the liver case.” Can you summarize those process more clearly as table?

P9, L286 The paragraph of “Earlier study showed BDT benefit for DDCS… be comparable between the clinical plan and the test plan, for both cases” should be combined in the section of 4.Discussion. 

P9, L295- P 10 Figure 4

The paragraph of “To understand why…. as compared with those of the corresponding clinical plans.” and figure 4 also should be moved to 4.Discussion. But it might be hard to discuss the effect of move dose contribution. The box plots shows there is no significant differences between the SR of clinical treatment plan and that of test plans. Please consider if the figure 4 is necessary in this manuscript or not.

P9, L298 “less move contribution” -> less move dose contribution? 

Round 2

Reviewer 1 Report

General Comments

This is the 1st version of a revised manuscript submitted to and reviewed for Cancers. The authors have satisfactorily addressed most of my concerns. I recommend its acceptance for publication.

Specific Comments

There are no specific comments on this version of the manuscript.

Author Response

We sincerely thank the reviewer for conducting a thorough review of our manuscript and providing invaluable comments that greatly contributed to its improvement. We are also grateful for the reviewer's approval of our revised manuscript.

Reviewer 3 Report

Please find review report attached.

Round 3

Reviewer 3 Report

I appreciate the authors’ efforts in revising the manuscript. All of my questions and comments have been adequately addressed.